# Geoparks and Geotourism in China: A Sustainable Approach to Geoheritage Conservation and Local Development—A Review

**Kejian Xu [1,*] and Wenhui Wu [2]**

[1]   School of Economics and Management, China University of Geosciences, Beijing 100083, China
[2]   Institute of Sedimentary Geology, Chengdu University of Technology, Chengdu 610059, China
*   Correspondence: xukj@cugb.edu.cn

**Abstract:** Geoparks and geotourism are relatively new activities within tourism. However, both have grown rapidly over the past decade. Geoparks, as an innovation for the conservation of geoheritage, play an important role in the development of geotourism. Geotourism has evolved partially in response to the need to minimize the negative impacts of mass tourism in geologically and geographically sensitive and/or im-portant areas situated in tourist environments, while at the same time providing a catalyst for sustainable rural development. China, with its vast territory and complex geological and geomorphic features, is often referred to as an open laboratory in geosciences and has 289 national geoparks and 41 UNESCO global geoparks so far. Currently, it is a leading country in the world in establishing and maintaining geoparks. This paper reviews the geoparks initiatives in China, as well as attempts to assess the compatibility of geoconservation and rural development within geotourism areas by exploring the challenges and outcomes of the geotourism development in China and by identifying and analysing the outcomes of geopark development. The results indicate a geopark is a sustainable approach to advancing geoconservation and promoting local economic development. The results further emphasize the importance of sustainable management in geotourism. Only when managed in a sustainable manner is geotourism likely to provide long-term improvements for developments in rural areas. Implications for geopark management and geotourism development are discussed.

**Keywords:** geoheritage; geoconservation; geopark; geotourism; sustainable development

## 1. Introduction

Geoparks and geotourism are two of the newest endeavours within tourism during the past decade and have become widely known [1–3]. Geotourism has developed to address the need to minimize the negative impacts of mass tourism at tourist sites based around geological and geomorphological attractions. Its key goal is an emphasis on sustainable tourism in primarily rural and natural environments [4]. A geopark is an area for sustainable development and is a global marketing concept. The United Nations Educational, Scientific and Cultural Organization (UNESCO) defined geoparks as nationally protected areas with a number of geoheritage sites of particular importance, rarity or aesthetic appeal [5]. The establishment of geoparks also favours non-traditional economic development based on geological features and geotourism [6–8]. The essential purpose of a geopark can be seen as establishing a natural history system which consists of geoheritage and other local value in its territory, in order to link heritage conservation with educational and economic development and, further, as a backup for geoconservation by utilizing educational and other local sustainable development activities. For this reason, both geoparks and geotourism may be seen as attractive endeavours for rural development in many peripheral areas facing emigration [1].

The idea of geoparks was developed in 1996 but it was only in 2000 that representatives of four European territories, which had separately been promoting geological conservation

and sustainable development, signed a convention declaring the creation of the European Geoparks Network (EGN) [9–11]. In 2004, the EGN was opened to non-European territories including eight Chinese geoparks, creating the Global Geoparks Network (GGN). The Madonie Declaration was also signed in 2004 re-affirming the previous agreement (2001) of cooperation between the Division of Earth Sciences of UNESCO and the GGN [11,12]. This cooperation was consolidated in 2015 when the 195 member states of UNESCO ratified the creation of the UNESCO Geosciences and Geoparks Programme, with the UNESCO Global Geoparks (UGGps) as a branch of this programme [13].

In April 2022, UNESCO's Executive Board approved the designation of eight new UGGps in the world. The total number of sites in the GGN is 177 in 46 countries. The distribution of UGGps is variable depending on the continents and countries (Figure 1). The great majority of UGGps are located in Europe (94 UGGps) and in the Asia-Pacific region (66 UGGps). There are eight UGGps in North America, seven UGGps in South and Central America and the Caribbean and two UGGps in Africa. China has 41 UGGps and it is the first country when considering the number of UGGps in the world.

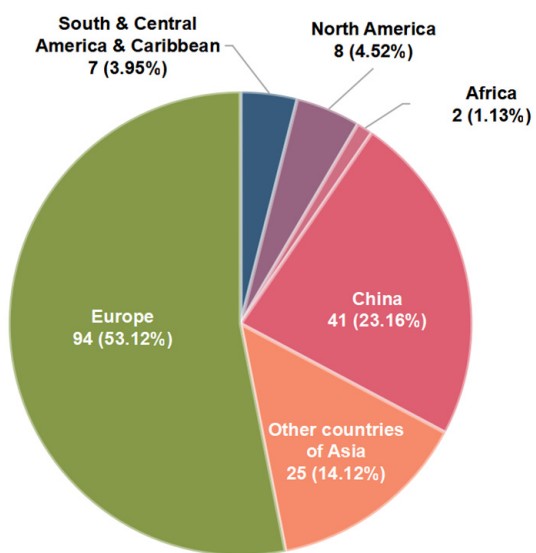

**Figure 1.** Distribution of the number of UGGps by continent.

China has a unique geology and a dynamic natural landscape which has attracted numerous tourists from home and abroad. Owing to the dynamic processes still shaping China's landscape, combined with the natural environment in the country, China is often referred to as an open laboratory in geosciences. From the 1990s, more and more attention has been gradually focused on geoconservation on account of incremental interest in and the establishment of geoparks and geotourism. The establishment of geoparks in China provides an optimal opportunity to achieve the goal of geoconservation while fostering sustainable development of rural areas, as its geological heritage is irreplaceable and vulnerable, and sustainable development is the only option for long-term development [14–16].

## 2. Literature Review

### 2.1. Geopark, Geoheritage and Conservation

A geopark is a territory with a particular (geologically-emphasized) heritage of international significance, though only from a geological standpoint; it may also be significant for archaeological, ecological, economical, historical and cultural reasons. In a geopark, all of these aspects should be linked in a sustainable territorial development strategy [17]. The promotion of sustainable development is thus a key aim of the geopark ideology. This is highlighted by Dowling (2010), who stresses that geoparks are primarily established to create enhanced employment opportunities for the local people as well as fostering their economic benefit. A geopark achieves its goal through a three-pronged approach:

geoconservation, geo-education and geotourism. Through geoconservation, a geopark seeks to conserve significant geological features, explore and demonstrate methods for excellence in conservation, and provide a venue for geo-education. Through geo-education, a geopark organizes activities and provides logistical support to communicate geoscientific knowledge and environmental concepts to the public through various means. With regard to geotourism, geoparks stimulate economic activity and sustainable development through geotourism and local commercial endeavours (restaurants, souvenirs, books and pamphlets). Most importantly, they encourage the participation of local communities in the creation of enterprises and service industries involved in geotourism and geoproducts through tourism planning, development and management [1].

Geoheritage is defined as the unique geosites/geological sites that are outstanding and representative in the area. Moreover, they should have significant geodiversity including scientific, educational, aesthetic, recreational, cultural, and other value [18–21]. Because of the value of geoheritage, they are essential for society and should be conserved for subsequent generations.

Geoheritage studies around the world have shown a marked growth since the early 1990s [22–26]. In addition, the International Union for Conservation of Nature (IUCN) and the United Nations Sustainable Development Goals (SDG) identify the importance of geoheritage and geoconservation [27,28]. Communities need to appreciate local geoheritage through tourism. Geotourism focuses on some aspects of geological and/or geomorphological heritage of the Earth that can benefit (or negatively affect) the geoheritage values of a region [29–31]. Once the geoheritage is recognized and valued, the need for geoconservation begins. The geoconservation that a community must work towards will ensure that conservation of geoheritage features in protected areas gains more popularity on local, national, and international levels for nature, sustainable development, and human well-being [32].

Over the past three decades, several quantitative and qualitative methods have proposed how to classify geoheritage and geodiversity. In 1993, the International Union of Geological Sciences proposed a classification scheme for geoheritage, including palaeontology, physiognomy, rock, strata, mineral, plate tectonics and submarine geomorphology [33]. The Geoparks Secretariat provides a classification scheme, with emphasis on the geoheritage features evident in various geoscientific disciplines such as: solid Earth sciences, economic geology and mining, engineering geology, geomorphology, glacial geology, physical geography, hydrology, mineralogy, palaeontology, petrology, sedimentology, soil science, speleology, stratigraphy, structural geology, volcanology, etc. [34]. To date, there have been a few different geoheritage classification schemes developed [35–38]. The classifications are similar to each other, mostly from the disciplines of geoheritage or the geological processes formed. Qi et al. (2004) proposed a comprehensive classification method based on 'materials composition and cause of formation' [39]. Xu (2007) divided geoheritage into 10 categories based on the origin of sections, strata, structures, rocks, fossils and topographies [40]. These classifications are superficial to a certain extent when geodiversity is considered. Therefore, an improved scientific and detailed classification method should be created.

Geoconservation is 'the dynamic preservation and maintenance of various geoheritage sites' [41]. Burek and Prosser (2008) defined geoconservation as 'the action taken with the intent of conserving geological features, processes, sites, and specimens' [42]. Geoconservation is currently defined as 'the act of protecting geosites and geomorphosites from damage, deterioration, or loss through the implementation of protection and management measures' [43]. Geoconservation, now a growing activity, given a general specification by Prosser (2013), refers to the behaviour aiming at implementing conservation for biotic and abiotic objects, including science education, scientific research, necessary data collection, and so on [44].

*2.2. Geotourism*

The concept and endeavour of geotourism is dynamic and continues to be redefined and refined [2,32,45–49]. One of the most recent definitions is given by Newsome and Dowling (2010) defining geotourism as a form of natural area tourism that specifically focuses on landscape and geology. It promotes tourism to geosites and the conservation of geodiversity and understanding of Earth sciences through appreciation and learning. This is achieved through independent visits to geological features, use of geotrails and viewpoints, guided tours, geo-activities, and patronage of geosite visitor centres [2]. With the rapid creation of geoparks worldwide, geotourism is gradually growing into a flourishing domain [50]. Hose (2011) viewed geotourism as an important vehicle for geological protection and conservation [48]. Geotourism also flourishes as a significant source of income through sustainable economic development, especially in underdeveloped rural districts, e.g., a new source of revenue which is generated through geotourism from cultural experience [45].

In China, geotourism is known as tourism Earth-science which uses the principles and methods of geology and tourism to study geotourism resources produced by geological processes, and to discover geological landscapes with ornamental, interesting and scientific value. It was put forward by Chen Anze and other Chinese scholars and has been extensively used in China for over thirty years [51–53]. With the initiative and development of geoparks in China, the theory of geotourism has become a new study field in the 21st century [54].

In recent years, geotourism research has mainly focused on the classification, distribution, assessment, development and conservation of geotourism resources, the coordinated development model of geotourism resources, and the relationship between mankind and the land [55–68]. There are also studies aiming at international comparisons, taking geotourism development of mature American national parks as a reference, and proposing methods suitable for Chinese geotourism development [69,70].

In the study of geoparks, geoheritage and geoconservation, foreign scholars pay more attention to geoconservation and the establishment and management of geoparks. For example, Rohling (2004) studied the aesthetic characteristics of geoheritage and explained the significance of geoparks to the public [71]. Eder and Patzak (2004) propose that the geopark is a tool for public education, entertainment and local economic development, and it is emphasized that geoparks should play the role of promoting local economic development [72]. Zouros and Martini (2001) studied the relationship between European geoparks, geoconservation, geotourism and the sustainable development of the local economy [10]. Some scholars in China consider related studies on geoheritage and geoparks as important [73–77]. These authors have attempted to extend such studies to include the geological background of national geoparks [75–78], geoheritage assessment [79–82], geoconservation [83–86], geoheritage development [86–88], relationship between geoheritage conservation and economic development [89–94], and the sustainable development of geoparks [95–98].

According to our literature review, we find most studies focus on geoheritage and geoconservation, and few papers discuss geoparks in a comprehensive way. We will explore two hypotheses in this study:

**Hypothesis 1:** *Geoparks have established long-term protected areas for geoheritage, and platforms for geoscience education.*

**Hypothesis 2:** *The coordinated development of geoparks and local communities is the ultimate goal of geoparks, which is helpful for the sustainable development of geoparks.*

The objectives of this study are to (1) describe the historical development and the status of the geoparks in China with an analysis of geological and geographical setting; (2) classify geoheritage in China; (3) assess the compatibility of geoconservation and rural development within geotourism by identifying and analysing the outcomes of geopark

development in China; and (4) discuss the main challenges of the future of geoparks in China. They are explored through the following research questions:

(1) What is the relationship between geoparks and geoconservation?
(2) Do geoparks promote the sustainable social and economic development of the regions and communities through geotourism?
(3) What are the policy implications of the management and development of geoparks in China?

## 3. Methods

To achieve the objectives of this study, we collected relevant published references through searching by key words such as 'geopark','geoheritage', 'geoconservation', and 'geotourism' from 1990 to the present and using information from the websites such as http://www.globalgeopark.org.cn/ (accessed on 30 June 2022), https://asiapacificgeoparks.org/ (accessed on 30 June 2022), https://www.unesco.org/en/search?category=Unesco.org&text=geopark (accessed on 30 June 2022), etc. Note that Brocx and Semeniuk (2007) provide a history of the inception and usage of the terms geoheritage, geoconservation, and geodiversity up to 2007 [21]. We gathered data from reports and documents from the MLR, the National Forestry and Grassland Administration, and Chinese Geoparks Network. We also incorporated the available annual detailed reports of Chinese geoparks and information provided on their websites. They include the data referring to visitors, jobs, tourism revenue, geoconservation funds, the cartographic data of China's geoparks, and the information on essential aspects to understand the geoconservation, management and geotourism in China's geoparks. Based on the previous methods of geoheritage classification and field investigations carried out in various geoparks in China during the last twenty years, we improved the previous classification scheme and reclassified the geoheritage in China. Through geopark managers' interviews, policy implications of the management and development of geoparks in China were provided (Figure 2).

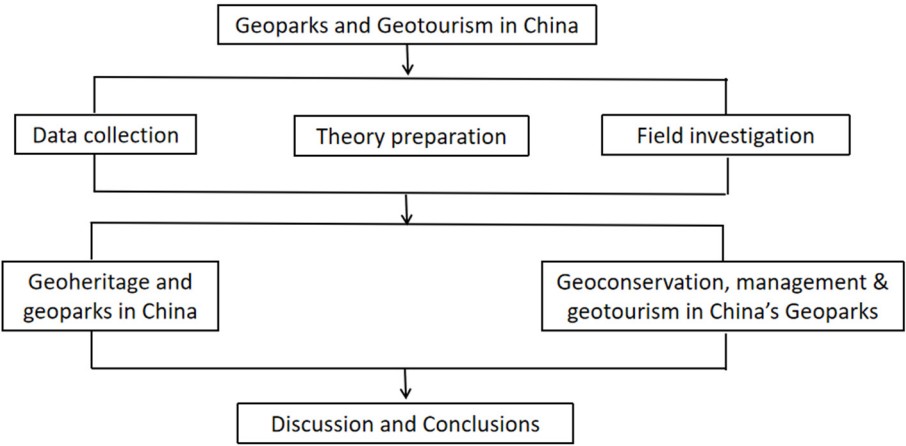

**Figure 2.** Flow chart of the research framework.

## 4. Geoheritage and Geoparks in China

### 4.1. Geological and Geographical Setting of China

China is located at the convergence of three geological plates: the Pacific Ocean Plate, the Eurasia Plate and the India Plate. In its long geological history, it has undergone the evolution of ancient plates such as Siberia, North China Tarim, South China, Yunnan-Tibet and India (Gondwana), therefore it has a complex and diverse geological and tectonic pattern (Figure 3) [99]. Many previous studies [100–102] have shown that the major fault and fracture systems in the region exert a dominant control on landform development, via differential uplift and widening of fractures and joint sets. The basement structure of most of southern China was established by the Wuling-Xuefeng Orogenic Movement (~1400–850 Ma) and, during the Late Triassic Indonesian Movement, China underwent

a complex process of NW-SE tectonic extension and EW shearing, producing associated groups of NW-NNW tension-faults and NS and NNW strike-slip faults. The early Yanshanian Movement during the late Jurassic period (around 160–140 Ma) caused EW extension of China and formation of NNE-NS folds together with two sets of strike-slip faults in NNW and NEE directions. The late Yanshanian Movement of the late Cretaceous age (around 130~68 Ma), displaced south China from SE to NW with reactivation and enlargement of faults, uplift and tilting [100].

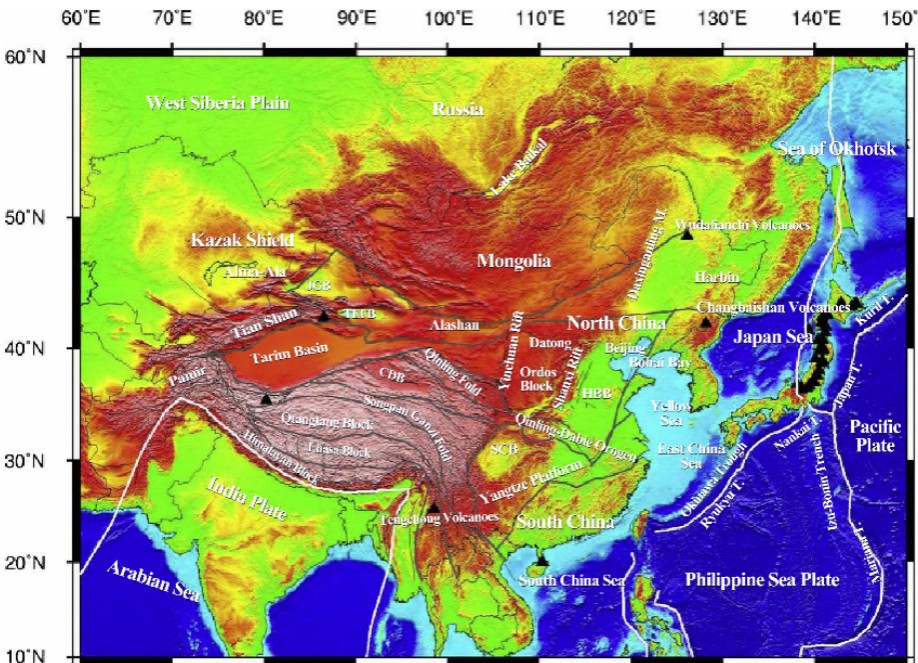

**Figure 3.** The geological setting of China (modified from [99]).

China has a well-developed stratigraphic history (Figure 4) [103], as well as various sedimentary rock types and complex geological structures where the active zones and stable zones co-exist. It also has experienced frequent magmatic activity, a long geological evolutionary history, as well as different styles and types of metamorphism. Therefore, globally, China is one of the regions with significant geological characteristics, where diverse geoheritage features and landscapes have been well-preserved.

China has an area of 9,600,000 km$^2$ and a coastline of 18,000 km, and is the world's third largest country, after Russia and Canada. It spans a latitude of about 50 degrees and a longitude of about 62 degrees. The vast land expanses of China include plateaux, plains, basins, hills, and mountains. They occupy nearly two-thirds of the land, higher in the west and lower in the east, which results in three tectono-geomorphic steps shown in Figure 5. The average altitude of the first step is more than 4000 m. The highest step is formed by the Qinghai-Tibet Plateau at an average height of over 4000 m, with the Kunlun Mountain Range, the Qilian Mountain Range and the Hengduan Mountain Range as the division between this step and the second one. The highest peak in the world, Everest, at 8844.43 m high is known as 'the Roof of the World'. On the second step are large basins and plateaux, most of which are 1000–2000 m in altitude. The Great Khingan, Taihang Mountain, Wu Mountain and Xuefeng Mountain divide this step and the next lower one. Plateaux including the Inner Mongolian, Loess, and Yunnan-Guizhou Plateaux, and basins such as Tarim, Junggar, and Sichuan Basins are situated here. The third step, easternmost and lowest, which has extensive broad plains, is dotted with hills and lower mountains, with altitudes of over 500 m. Here are located famous plains: the Northeast Plain, the North China Plain, and the Middle-Lower Yangtze Plain, neighbouring each other from north to south.

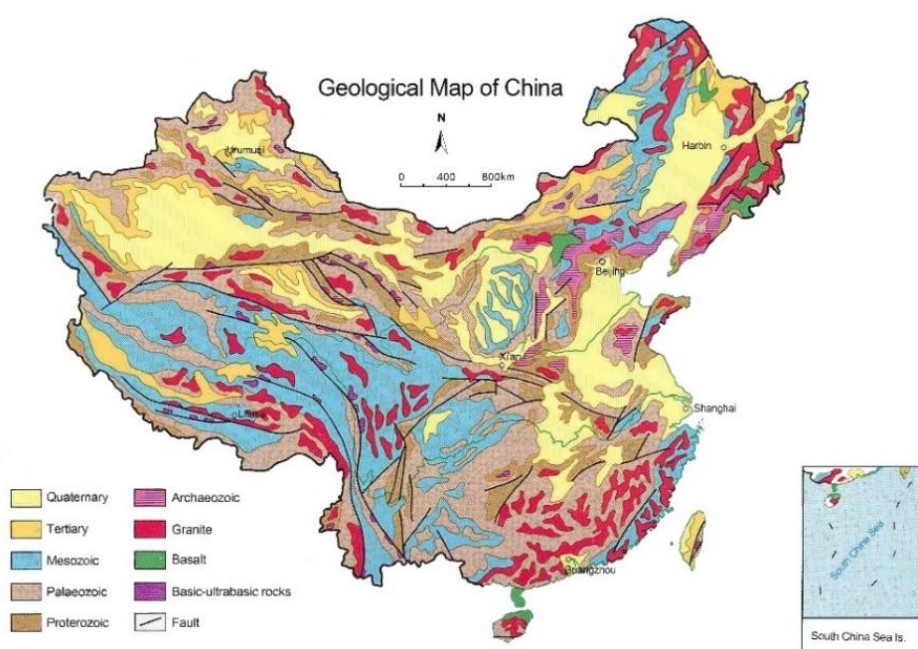

**Figure 4.** Geological map of China (modified from [103]).

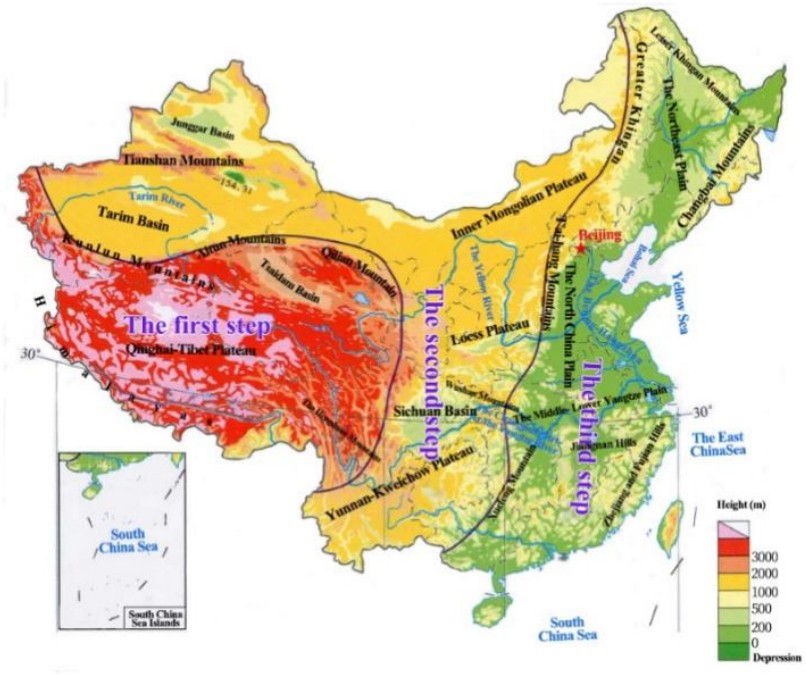

**Figure 5.** Three Tectono-Geomorphic Steps (modified from [104]).

*4.2. Classification of Geoheritage in China*

China possesses numerous outstanding and varied geoheritage features owing to the three tectono-geomorphic steps. However, until the 1980s there were no serious efforts in China to conserve sites of geoheritage significance from unsustainable resource development and utilization [98]. Since then, more attention has been gradually focused on geoconservation on account of incremental interest in geoheritage. Based on previous work and the programme 'technical requirements of China's national geoparks' released by the MLR, geoheritage in China can be classified into three different categories, i.e., basic geology, landforms, geohazards, and 15 subcategories (Table 1).

**Table 1.** The classification of geoheritage in China in UGGp.

| Category | Subcategory | Geoheritage | Representative Geoparks |
|---|---|---|---|
| basic geology | stratigraphic section | international stratigraphic section | Songshan UGGp, Wangwushan-Daimeishan UGGp, Xiangxi UGGp, Shennongjia UGGp |
| | | stratigraphic (typical) section | |
| | | geological events section | |
| | rock section | intrusive rocks section | Taishan UGGp, Tianzhushan UGGp, Keketuohai UGGp, Yimengshan UGGp, Huanggang Dabieshan UGGp, Dali-Cangshan UGGp |
| | | volcanic rocks section | |
| | | metamorphic rocks section | |
| | | sedimentary rocks section | |
| | structural section | plane of unconformity | Qinling Zhongnanshan UGGp, Tianzhushan UGGp, Lushan UGGp, Fangshan UGGp, Yanqing UGGp, Wangwushan-Daimeishan UGGp |
| | | fold and deformation | |
| | | fault | |
| | important fossil site | palaeoanthropolgy fossil site | Fangshan UGGp, Zigong UGGp, Funiushan UGGp, Tianzhushan UGGp, Yanqing UGGp, Shennongjia UGGp |
| | | palaeontology fossil site | |
| | | palaeophytology fossil site | |
| | | palaeozoology fossil site | |
| | | ichnofossil site | |
| landforms | loess and rock landform | carbonate rock landform | Shilin UGGp, Xingwen UGGp, Leye-Fengshan UGGp, Fangshan UGGp, Zhijindong UGGp, Guangwushan-Nuoshuihe UGGp |
| | | granite landform | Huangshan UGGp, Sanqingshan UGGp, Tianzhushan UGGp, |
| | | clastic rock landform | Danxiashan UGGp, Zhangjiajie UGGp, Longhushan UGGp, Dunhuang UGGp, Yuntaishan UGGp, Yimengshan UGGp, Taining UGGp |
| | | loess landform | Luochuan National Geopark |
| | dessert landform | desert landform | Alxa Desert UGGp, Hexigten UGGp, Dunhuang UGGp |
| | | gobi landform | |
| | volcanic landform | volcanic apparatus | Wudalianchi UGGp, Jingpohu UGGp, Leiqiong UGGp, Yandangshan UGGp, Hexigten UGGp, Hong Kong UGGp, Arxan UGGp |
| | | volcanic lava landform | |
| | | volcaniclastic accumulation landform | |
| | glacier landform | glacier erosion landform glaciers accumulate landform | Kunlunshan UGGp, Dali-Cangshan UGGp |
| | ice-marginal landform | ice-marginal deposit | Kunlunshan UGGp, Shennongjia UGGp |
| | | ice-marginal structure | |
| | | ice-marginal terrain | |
| | coastal landform | marine-erosion landform | Hong Kong UGGp, Leiqiong UGGp |
| | | marine-deposit landform | |
| | fluvial landform | fluvial erosion landform | Fangshan UGGp, Ningde UGGp, Shennongjia UGGp, Dali-Cangshan UGGp |
| | | fluvial deposit landform | |
| | structural landform | tectonic structure landforms | Taishan UGGp, Fangshan UGGp, Yanqing UGGp, Yuntaishan UGGp |
| | | mesoscopic structure landform | |
| | | minor-scale structure landform | |

| Category | Subcategory | Geoheritage | Representative Geoparks |
|---|---|---|---|
| | | river landscape | Fangshan UGGp, Hexigten UGGp, Ningde UGGp |
| | | lake, pool | Jingpohu UGGp, Dali-Cangshan UGGp, Wudalianchi UGGp |
| | water landscape | wetland-marsh | Dali-Cangshan UGGp, Hexigten UGGp |
| | | waterfall | Yuntaishan UGGp, Yandangshan UGGp, Jingpohu UGGp |
| | | spring | Wudalianchi UGGp, Arxan UGGp, Dunhuang UGGp |
| | earthquake | geosuture | Kunlunshan UGGp, Keketuohai UGGp |
| | | ground deformation | |
| geohazards | | collapse | |
| | | landslide | |
| | geohazard remains | debris flow | Qinling Zhongnannshan UGGp |
| | | surface collapse | |
| | | ground subsidence | |

### 4.3. The Initiative of Geoparks in China

In China, the concept and management of geoparks were introduced in the early 1990s. Encouraged by the UNESCO Geoparks programme, the Ministry of Geology and Mineral Resources officially accepted the idea of geoparks, subsequently, the MLR issued a notification to approve national geoparks and outlined the regulations regarding the evaluation criteria, and approval procedures in 2000, opening a way for the central government to manage national geoparks. In 2002, the Chinese Geoparks Network (CGN) responsible for the coordination and management of national geoparks and global geoparks was established. From 2003, in response to the establishment of the GGN, China's geoparks began to apply for global geopark designation. When 25 global geoparks in the world were announced in 2004, eight of them were located in China, demonstrating that China's geoparks were a success. Due to reform by the Chinese government, in 2018, the newly founded National Forestry and Grassland Administration of PRC was made responsible for the geoparks administration instead of the MLR. At present, there are 41 UGGps and 281 national geoparks in China, covering an area of over 60,000 km$^2$. (Figures 6 and 7). The spatial distribution of Chinese geoparks coincides with the Chinese geological structure features of three-step topography. Various typical geoparks are distributed in ladders or transitional zones.

China has been very active in promoting geoparks so as to conserve geoheritage and raise public awareness about geological and environmental protection. This can be seen from past conferences and practices in which China was positively involved and played an important role. The first International Conference on Geoparks, that was then held in June 2004 in Beijing, China, was a historical event which conveyed a message to the rest of the world that China has a strong will in the promotion of the concept of geoparks.

The establishment of the GGN was a good sign from the perspective of providing an efficient method of local economic development as well as geological conservation. It encouraged many Chinese national geoparks in becoming aspiring UNESCO Global Geoparks, and then UNESCO Global Geoparks. As shown in Figure 8, after the eight national geoparks initially joined the GGN in 2004, four other Chinese national geoparks were ratified as GGN members in 2005, with six new members of the GGN in the year of 2006 being from China. Between the years 2008–2015, almost every year the GGN welcomed two new Chinese members, with the exception of 2012, in which only one joined

the GGN. It is also noticeable that there were two more new Chinese members in each year after 2017. China's continuous contribution to the GGN is certainly derived from the rapid development of UNESCO global geoparks in China.

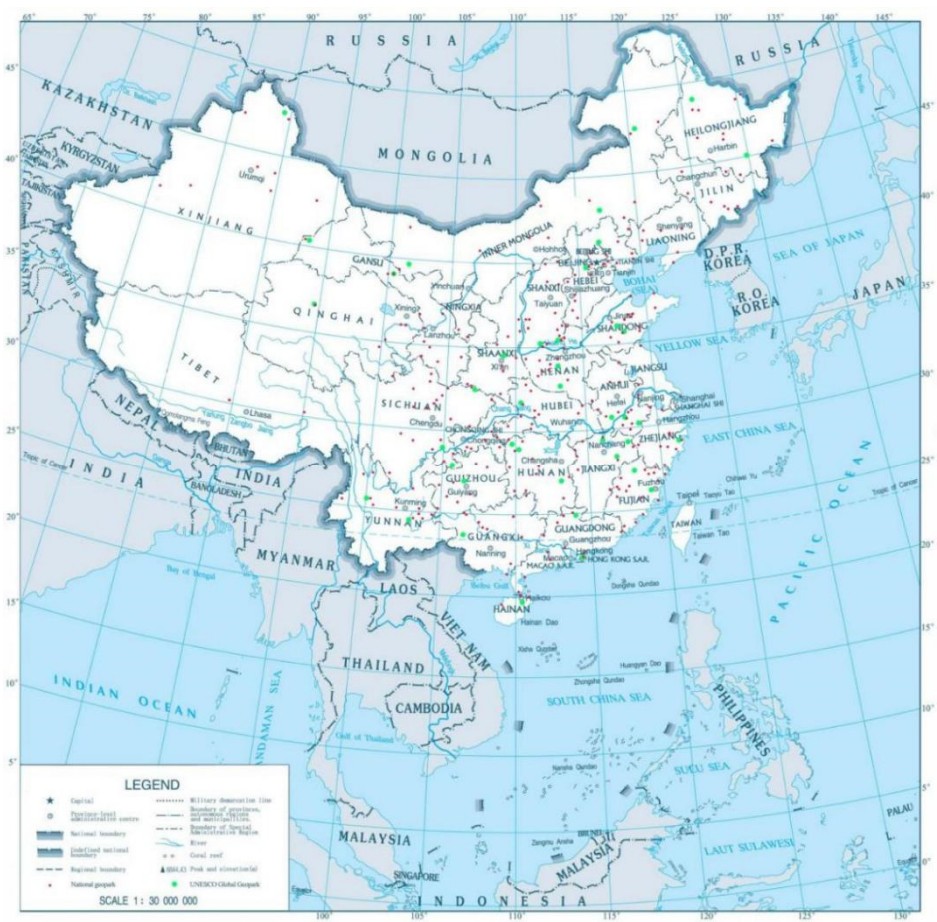

**Figure 6.** Distribution of Geoparks in China.

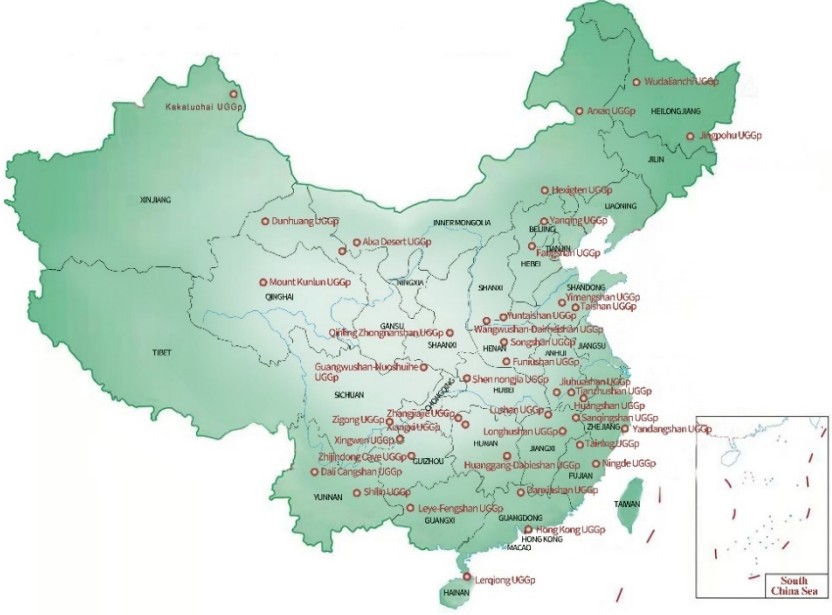

**Figure 7.** Map of UNESCO Global Geoparks in China.

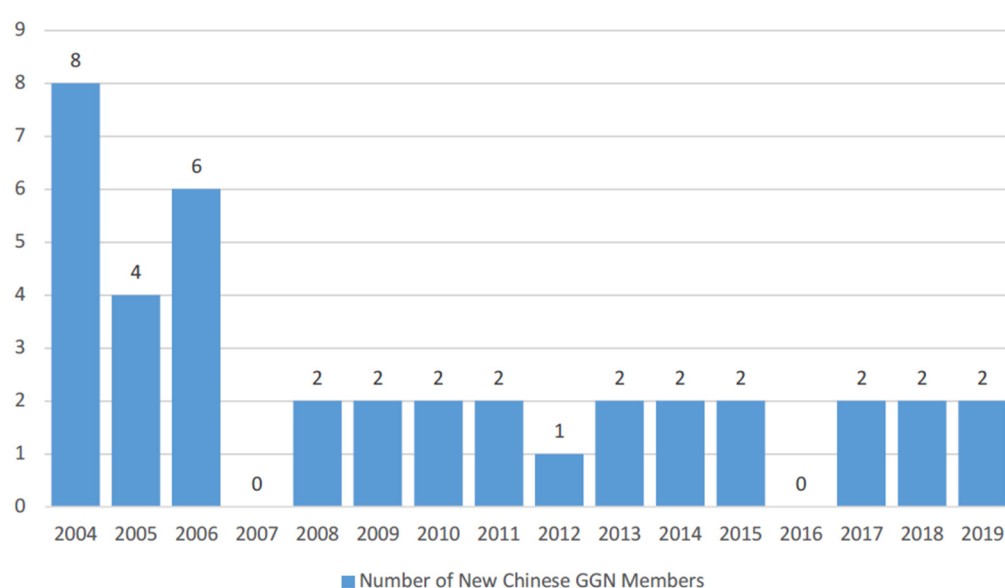

**Figure 8.** Number of New GGN members from China (2004–2019).

## 5. Geoconservation, Management, and Geotourism in China's Geoparks

*5.1. Geoconservation*

### 5.1.1. Conservation Legal Framework

China's geoparks including 41 UGGps and 281 national geoparks are State-owned assets. They protect geoheritage. All national laws and regulations are the legal basis for geoconservation. There are no specific laws on UGGps in China, and drafts of regulations on national geoparks have been completed for many years but have not been ratified However, the State Council has set forth administrative regulations, including Regulations on Scenic and Historic Areas (2016 Revision), Regulations on the Protection of Geoheritage (1995), Regulations of the People's Republic of China on Nature Reserves (2017 Amendment), Regulations on the Protection of Fossils (2010), Forest Law of the People's Republic of China (2009 Amendment), Environmental Protection Law of the People's Republic of China (2014 Revision), and Mineral Resources Law of the People's Republic of China (2009 Amendment). Regarding Regulations on the Protection of Geoheritage, according to the provision of law, the Environmental Protection Department under the State Council should provide assistance, and the geology and mineral resources department under the State Council should supervise the protection of geoheritage sites nationwide.

### 5.1.2. Zoning for Geoconservation

A three-tier protection zoning system has been adopted in China's geoparks, i.e., core protected areas, special protected areas, integrated protected areas and is shown in Table 2. The core protected areas are places where the geoheritage is of international or national scientific significance which have been preserved in their natural state and are extremely sensitive to human impact. These places have been designated mainly for conservation purposes, where no infrastructure is permitted. Anyone without permission is prohibited from entering. The special protected areas are places where geoheritage is of regional scientific significance and has medium carrying capacity and sensitivity. Scientific research, educational activities, academic exchanges and geotourism activities are allowed. The integrated protected areas have high carrying capacity. Most tourism facilities are already in place and the areas can serve recreational purposes.

**Table 2.** The three-tier protection zoning system of China's geoparks.

| Protection Level | Vulnerability | Carrying Capacity | Naturalness | Safety Level |
|---|---|---|---|---|
| Core protected area | High | Low | High | Low |
| Special protected area | Medium | Medium | Medium | Medium |
| Integrated protected area | Low | High | Medium | High |

*5.2. Management*

5.2.1. Geoparks Management Structure

China's geoparks are established through a top-down process and have a multi-tier administrative structure. The top tier is the National Forestry and Grassland Administration, the second is the provincial Department of Forestry and Grassland, the third is the municipal people's government of Forestry and Grassland, the fourth is the geopark administrative committee, and the fifth is the geopark management bureau, which is responsible for geoparks (Figure 9). The government has a crucial role in decision-making and management of the geoparks, which is most likely the main difference between Chinese geoparks and any other geopark in the world.

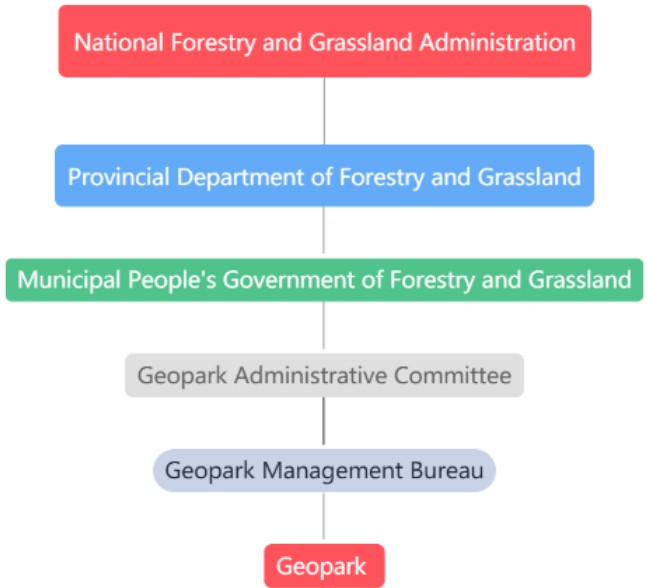

**Figure 9.** Chinese Geoparks Management Structure.

5.2.2. Geoparks Management Body

There are three types of geopark management bodies in China. The first is an independent management body. It is responsible for the implementation of government projects; geoheritage survey and assessment; the conservation of the geological heritage and ecological environment; the organization and implementation of scientific research; science education; marketing and promotion, infrastructure construction, and exchange with other geoparks. The second type is a shared management body. It is used to administer geoparks and other designations (world heritage site, world biosphere reserve, AAAAA national tourist attraction, national park, national forest park, nature reserve of China, China cultural heritage, etc.) and is under the leadership of the local government's administration committee (Table 3). It has its own office, which is only responsible for the daily work of the geopark, e.g., Huangshan UGGp, Lushan UGGp, Taishan UGGp, Sanqingshan UGGp, Yandangshan UGGp, Shennongjia UGGp, Dunhuang UGGp. China's geoparks typically cover a broad range of areas and overlap with other protected areas or national brand tourism destinations. Therefore, a shared management body facilitates the planning, conservation, and management of the area. For instance, Figure 10 shows that

the Dunhuang UGGp overlaps with many other brands of geoparks in China but is under the leadership of the Dunhuang Municipal Government (which administers other tourism areas in Dunhuang City). The third type is a management organization subordinate to a government department. For example, Xingwen UGGp is subordinate to the local natural resources and planning bureau. The administration of Danxiashan UGGp is subordinate to the local tourist administration. Although subordinate to government departments, the administration is still responsible for the management and geoconservation of geoparks, and associated geoeducation.

**Table 3.** Other designations in addition to UGGps (modified from [105]).

| Name | UNESCO Designation | | | | National Designation | | | |
|---|---|---|---|---|---|---|---|---|
| | UNESCO Global Geopark | World Heritage Site | World Biosphere Reserve | AAAAA National Tourist Attraction | National Park of China | National Forest Park of China | Nature Reserve of China | China Cultural Heritage |
| Huangshan | ✓ | ● (Mixed) | ✓ | ✓ | ✓ | | | ✓ |
| Lushan | ✓ | ● (Cultural) | | ✓ | ✓ | | ✓ | ✓ |
| Taishan | ✓ | ● (Mixed) | | ✓ | ✓ | | | ✓ |
| Sanqingshan | ✓ | ● (Natural) | | ✓ | ✓ | | | ✓ |
| Yangdangshan | ✓ | | | ✓ | ✓ | ✓ | | |
| Shennongjia | ✓ | ● (Natural) | ✓ | | | | ✓ | |
| Dunhuang | ✓ | ● (Cultural) | | ✓ | ✓ | | ✓ | ✓ |

Category of World Heritage Site: ● Cultural site, ● Natural site, ● Mixed site.

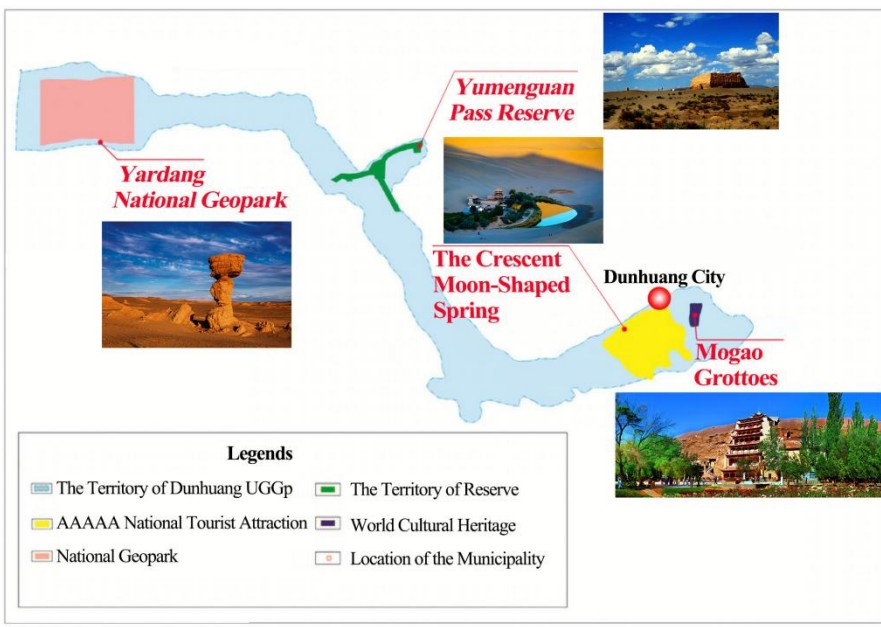

**Figure 10.** The Dunhuang UGGp overlaps with many other brands (modified from [105]).

5.2.3. Geoparks Operating Mechanism

Under the supervision of local governments, the main operating mechanism of China's geoparks consists of two types of enterprises: State-owned enterprises and private enterprises.

- State-owned enterprises. Authorized by the local government, the State-owned enterprises manage all resources within the geopark in accordance with a market-oriented

approach; most importantly, they are responsible for geoconservation. Ticket income belongs to the administration committee (government); other business income belongs to the enterprise or the administration committee; and the enterprise might share ticket revenue; and

- Private enterprises. Entrusted by the government to operate all tourism businesses, including tickets, the private enterprise is the only agency responsible for tourism management in the geopark. Thus, monopolizing the resources in the geopark, the enterprise is required to pay a franchise fee to the geopark administration, which is agreed upon in advance [105].

5.2.4. Geoparks Financial Management

The main funding sources of China's geoparks are described as follows:

- Fiscal policy of local governments. For large investments in tourism infrastructure, the geopark administration submits a budget to the financial department of the government one year in advance. After the government examines and approves the budget, appropriation can be allocated as planned in the subsequent year;
- Special funds for geoconservation. Special funds are issued by the Ministry of Finance and the MLR or by the provincial finance department to protect geoheritage at the global and national levels. These funds are utilized for tasks such as project engineering, educational activities, specimen collection, exhibitions;
- Support from the China Geological Survey. Funding is provided for geoscientists in geoheritage surveys and related research. From 2002 to 2017, the central government invested 4 billion yuan in the special funds for geoconservation, thereby, it spurred local governments at all levels to invest 4.3 billion yuan and attracted 33.8 billion yuan of social funds [106];
- Operating income. As geoparks are State-owned assets, regardless of the management structure, the geopark administration committees belong to the government; thus, enterprises pay rent or management fees to the administration committee every year. Franchising projects and tickets serve as operating income shared by the administration committee; and
- Social investments. Enterprises invest in geoparks with operation projects. From 2002 to 2017, geoparks received 33.8 billion yuan of social funds [106].

*5.3. Geotourism*

Geoparks are pioneers in the development of geotourism; they generate new job opportunities and new economic and cultural activities, and they should be sustainable and produce real economic benefits to local populations.

5.3.1. New Job and Business Opportunities

Due to the development of geoparks in China, such new job and business opportunities as travel agents, guides, event organizers and operators, transportation, shops, restaurants and cafés, accommodation, infrastructure construction companies, museum design, printing, local food and souvenir manufacturers, environmental survey, planning and conservation professionals (geoscientists, ecologists, etc.), have been created. From 2002 to 2017, the number of direct and indirect employees in geoparks was between 464,100 and 25,855,000, respectively [106]. The number of hotels, motels, hostels and farm inns within geoparks has reached more than 23,500 [106].

5.3.2. Increasing Number of Visitors and Revenue Generation

Geoparks have stimulated socio-economic activities and sustainable development by attracting an increasing number of visitors. By 2015, China's geoparks received 438 million visitors and tourism revenue reached 149.279 billion yuan [106].

Geoparks are a driving force in some remote areas or places that are experiencing shifts in their demography and industries. For instance, the Zhijindong Cave UGGp is

located in the western Guizhou Plateau, a remote area of Southwest China; despite its unique natural charm and high aesthetic value, it remains relatively unknown compared to other karst landforms, although it has offered tourism activities since 1987. In 2015, after it became a member of the GGN, the number of visitors increased by 51% compared with the previous year [106].

Figure 11 shows the number of tourists and amount of tourism income of Sanqingshan UGGp from 2012 to 2020. The Sanqingshan Geopark was approved as a member of the GGN in 2012; the number of tourists gradually increased, and it received 1.89 million tourists and generated 22.038 billion yuan in 2019.

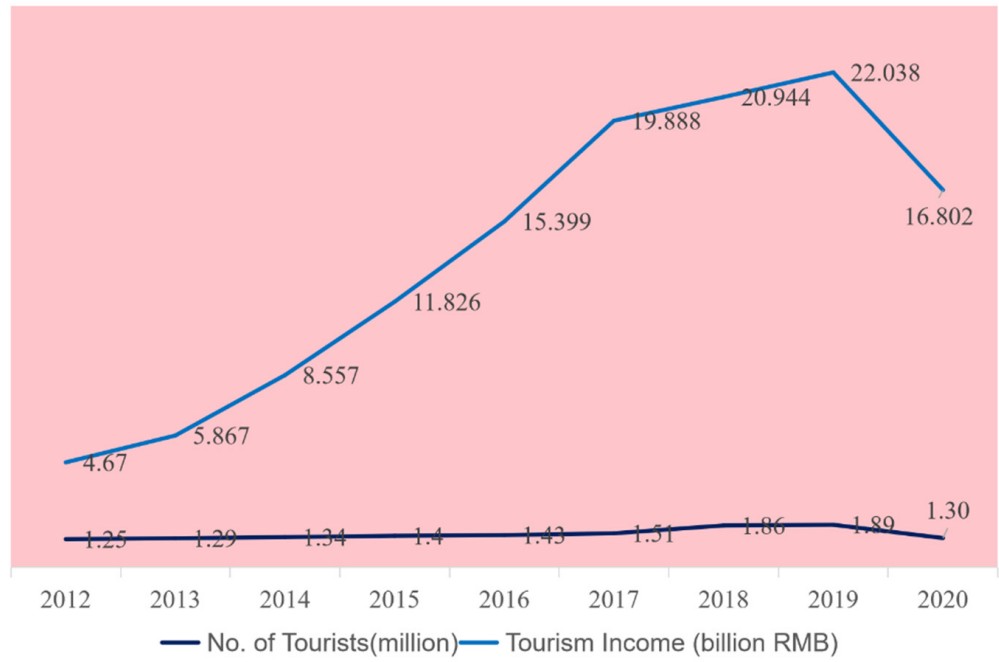

**Figure 11.** The number of tourists and amount of tourism income of Sanqingshan UGGp (2012–2020).

## 6. Discussion

### 6.1. Geoparks and Geotourism: Tools for Geoconservation and Local Development?

The establishment of geoparks presents a sound opportunity for China to achieve geoconservation of its resources whilst fostering sustainable development of its regional areas, as China's natural features are very vulnerable and sustainable use is the only option.

Geoconservation is a key element in geotourism development in China. Fundamental to the management of sustainable geotourism is an understanding of the interrelationship between geology, geodiversity, geoheritage, conservation, and tourism development, emphasizing the critical importance of research in order to find the ideal balance between these central components. However, geoparks do not have a positive reputation in the tourism market. In some traditional tourism areas or developed areas, due to overlap with other tourism brands, it is difficult to identify the difference between geotourism activities and traditional tourism activities. From the data of tourism, it is also impossible to distinguish the tourists' purpose and the reasons for the destination brand choice.

### 6.2. Visitor Management

In terms of protection, China has the largest population in the world but the worst vacation schedules. Geoparks will receive a large number of tourists during the peak season, which causes crowding. This large influx of tourists will not only place great pressure on the reception but also destroy the original and important geological and ecological environments. Therefore, restricting the flow of people is a common way to balance tourism and conservation in China.

For example, during the May Day holiday in 2021, Huangshan UGGp received a total of 104,997 tourists. Due to the limited visitor capacity of the geopark, to manage the increase in the number of tourists during the peak tourism season, the visitor flow in the geopark was effectively monitored to prevent accidents. Huangshan UGGp adjusts its tourist threshold according to the climate: the number of visitors in good weather is limited to 50,000, whereas the number of visitors in inclement weather (thunderstorms, strong winds, fog, and hail) is limited to 35,000 [105]. When the upper limit is reached, the system will stop selling tickets or encourage visitors who have bought tickets to visit the next day; through scientific calculation, the maximum number of visitors per day in Mogao Grottoes, Dunhuang UGGp, is strictly limited to less than 6000 [107].

At the same time, China mainly adopts engineering protection, manual inspections, video monitoring, and other means to protect features of geoheritage significance. For instance, visitors to Zhangye UGGp are made aware that the rainbow hills are not renewable so that deliberately climbing over a fence into a protected area and trampling it (which not only violated the law but also caused damage that could take a maximum of 60 years to recover) has legal consequences [105].

## 7. Conclusions

With its vast territory and diverse geological and geographical settings, China has 289 national geoparks and 41 UGGps so far, currently making it a leading country in the world in establishing and maintaining geoparks.

The present analysis has shown the main ideas to understanding the success of the implementation of geoparks in China relate not only to the number of geoparks, but also to their variety, breadth of representation, and number of activities.

This study proposes a classification system, and describes the geological background and geographical distribution of the national geoparks in China. The main conclusions are summarised as follows.

(1) Chinese geoparks are extensively distributed. Geoheritage sites in China are divided into three categories and 15 subcategories according to a new method based on the 'cause of formation'–protection attributes; and

(2) The spatial distribution of Chinese geoparks coincides with the Chinese geological structure features of a three-step topography (i.e., various typical geoparks are distributed in ladders or transitional zones).

The geopark and geotourism approach adopted in China is proven to be economical and effective in both geological conservation and rural development, especially in vulnerable environments. During the past two decades, the geoparks in China have made significant achievements in geoconservation and geotourism. However, some geoparks in China are still in the primary stage of development, and there are problems such as inefficient conservation management, the lack of geopark legislation, ineffective interpretation and educational activities, too little community involvement, etc.

China's geoparks need to improve in the following ways:

- Enhance the promotion of the sustainable development of geoparks:

In terms of benefits, the local governments with jurisdiction over some geoparks attach importance only to declarations or revalidation and achieving status as a global tourism brand, which is regarded as the definition of geopark success. Moreover, after receiving a UNESCO certification, some authorities no longer pay attention to the sustainable development and protection of geoparks as intended by the master plan. The lack of a master plan for sustainable development and protection can easily lead to tourism over-development, emphasizing a short-term economic value over protection. Regardless of the motivations for investing in geoparks in China, from the past to the future, the promotion of the sustainable development of geoparks, especially in terms of sustainable tourism, is a crucial task.

- Increase community participation:

Due to the top-down approach of China's geoparks, the administration of geoparks is actually an agency of the government and the government leads the whole process. Community participation is superficial, i.e., only at the economic level in some areas, such as geotourism, geo-restaurants, geosouvenirs, geo-food. Therefore, enthusiasm among residents should be encouraged and opportunities for public opinion should be increased, for example, establishing a council that includes people from different backgrounds so that their opinions can be heard during any decision-making process [108].

- Develop more geotrails and attract more geoguides:

Despite the increasingly large number of visitors in China, most visit as part of tour groups that are limited by time and trip mode with fixed routes and do not distinguish among ages or groups. Therefore, geoparks still focus on sightseeing tours. Although geotrails and some supplemental activities with extra fees exist, due to the high costs and limited quantities, these activities are not commonly experienced by tourists. Tour guides in China typically major in tourism rather than in geology-related subjects. Similarly, during the peak season, many part-time guides who are not familiar with the concepts of geology, geoheritage, and geoparks are hired by travel agencies. Therefore, it is crucial for the geoparks in China to develop more geotrails and attract more geoguides.

- Pay attention to the negative impact of tourism on the environment:

Geoparks with sensitive ecosystems and poor carrying capacities are facing a sustainability problem, especially some areas with glaciers or deserts, which pose the greatest difficulty in building and maintaining the infrastructures that are needed for tour groups. Firstly, due to the limited environmental carrying capacity, too many tourists will have a certain impact on the geology and ecology, which will affect the outcrops and the ecological balance and natural resources in the area. Secondly, due to the limited reception capacity, there are large numbers of tourists in the peak season, which causes crowding, which will not only accelerate damage to infrastructure but can also lead to a poor tourist experience.

**Author Contributions:** Conceptualization, K.X.; methodology, W.W. and K.X.; formal analysis, K.X. and W.W.; investigation, K.X. and W.W.; resources, K.X. and W.W.; data curation, K.X. and W.W.; writing—original draft preparation, K.X. and W.W.; writing—review and editing, K.X. and W.W.; visualization, K.X. and W.W.; supervision, K.X. and W.W.; project administration, K.X. All authors have read and agreed to the published version of the manuscript.

**Funding:** This research was funded by Fangshan UNESCO Global Geopark Administrative Committee (HBA150).

**Institutional Review Board Statement:** Not applicable.

**Informed Consent Statement:** Not applicable.

**Data Availability Statement:** Not applicable.

**Acknowledgments:** The authors would like to thank Joseph Finch for language proofreading and Wanli Wu for constructive comments and suggestions. We also thank Dunhuang UNESCO Global Geopark Administrative Committee for the photos in Figure 10 and Sanqingshan UNESCO Global Geopark Administrative Committee for the data in Figure 11.

**Conflicts of Interest:** The authors declare no conflict of interest.

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
