# Peer review of "Geoparks and Geotourism in China: A Sustainable Approach to Geoheritage Conservation and Local Development—A Review"

_land, doi:10.3390/land11091493_

Round 1

Reviewer 1 Report

This article appears to be well organized and structured and uses appropriate methodologies to discuss the main challenges of the future in China’s geoparks. I didn't notice any errors in the text, figures, or tables. The meaning of MLR is clarified only in chapter 3.3, although it was mentioned earlier in chapter 3.2.

Reviewer 2 Report

Dear authors,

Unfortunately, the article presented to me for review is quite interesting, but it is not a scientific article, but a popular science one. You have only provided a compilation of some geoparks in China, but what was the research problem, what hypotheses? The aim of the article is typically task-oriented, but not scientific. You have a good introduction, but think about what you want to discover as scientists.

Reviewer 3 Report

Abstract

                The abstract of the paper touches upon the main theme of the same, which is that of geo tourism in China. How it is that a sustainable approach can be adopted towards, geo heritage conservation as well as local heritage in the context of China, serves as the main question of the research paper, and this is mentioned in the abstract itself. The abstract provides mention of some basic definitions and concepts pertaining to geo heritage and conservatism and why it is that such a study needs to be conducted with reference to the country of China, is articulated in the abstract.

Introduction

            The introductory section of this research paper makes known the need of sustainable tourism in the country of China and how making use of geo tourism is something that can prove to be beneficial for the nation. The introduction makes a reference, to the fact that geo tours and geo parks are quite the fad among tourists in every part of the world today, and by making use of such attractions to lure tourists from around the world, China will not only be able to attract a lot of revenue, but it will also be able, to do so without harming the environment.

Literature Review

            The literature review is a section of the research paper that provides reference to a number of scholarly studies that have been conducted on the subject, of geo tourism over the years, including studies that have been carried out on China. The significance of sustainable tourism in the present day and age is a fact that is pointed out time and again in the review of literature, and the absence of sufficient research on the subject is highlighted in the literature review as well, the rationale for the study thus being well established.

Methodology

            A combination of primary and secondary research has been adopted for this study, with facts and figures on the subject of geo tourism in China, and the presence of Geo parks in this part of the world having been collated from a wide range of sources, and then synthesized and analyzed for the purpose of this study. Cartographical data is what constitutes the most important source of knowledge insofar as this study is concerned, and the findings generated by the analysis of this data is supported with the help of data that has been derived from secondary research.

Results of the Study

            The results of the research paper have been presented in quite a graphical form, with plenty of maps, graphs, and tables having been used to showcase the findings that were extracted from the study and the analysis of cartographical data. The results show that while China is doing a good job of implementing geo-tourism, there are more improvements that it can make to bring about a better way of doing this.

Conclusion

            The concluding section of the research paper is one that ties up all the crucial arguments that have been made throughout the same, with suggestions being made for more research to be taken up in this field, down the line. 

Comments fo authors

Point 1: The Introduction section could be improved. Authors must include more information about the research topic. For such a large subject, the presentation of the study area is brief and general.

Point 2: The very limited literature review should be expanded in accordance with the paper. I recommend including a section of the literature review that addresses this issue.

Point 3: The reference must be significant.

Point 4: Based on journal recommendations, the format should be improved.

Reviewer 4 Report

Although this manuscript is dealing with very important issue of geoparks and geotourism, the manuscript looks like a report rather than a scientific paper.

- Abstract is very generic and must be revised to include key findings. 

- Methodology section must be revised to clearly mention which kind of literature is included to conduct this study and show the whole methodology through a flowchart.

-Although result section is fine, still key findings should be represented more clearly in terms of quantitative terms.

- Bullet points should be removed from conclusion section.

For more comments, please find the reviewed manuscript attached herewith. 

Round 2

Reviewer 2 Report

Dear authors,

your article may be interesting, but you need to make some changes.

1. please specify - in the introduction you write that in China it was proposed to establish a geo-park in 1985, and in the following parts of your work it is mentioned the beginnings of the organization of geoparks in the early 1990s.

2. the research hypotheses presented in the paper, the aim of the paper should be a summary of the literature review, and not an introduction to the methodology. The text should not be broken down into chapters.

3. To Hypothesis 1. in your analysis, please show how many geoparks in China are protected by geo-heritage.

4. To hypothesis 2., please confirm what the economic benefits looked like before and after the establishment of the geopark, or how they increased in the years of the park's operation. You write yourself that not every park has yet recorded the expected economic increases.

5. Please detail your research methodology. Where did you look for information - on which websites, portals, etc., what was the main search word, from which period you analyzed the articles. What reports and documents were analyzed by you, what information did you find there.

6. Please add information, and maybe it will come out of your analysis because geoparks in China have been created, what is protected in them, or what we want to show (mountains, volcanoes - how many parks in which).

Reviewer 4 Report

Although this revised manuscript looks much better, methodology section still needs to be revised. Especially I still don't find the information about the source and screening methodology of  retrieved articles used for this study. 
